# Relationship between Body Composition and Performance Profile Characteristics in Female Futsal Players

Mónica Castillo [1], José Miguel Martínez-Sanz [1,*], Alfonso Penichet-Tomás [2], Sergio Sellés [2], Estela González-Rodriguez [1], José Antonio Hurtado-Sánchez [1] and Isabel Sospedra [1]

[1] Nursing Department, Research Group on Food and Nutrition (ALINUT), Faculty of Health Sciences, University of Alicante, 03690 Alicante, Spain

[2] Department of General Didactics and Specific Didactics, Faculty of Education, University of Alicante, 03690 Alicante, Spain

[*] Correspondence: josemiguel.ms@ua.es

**Featured Application: The results presented in this study provide a reference point on the most important anthropometric determinants of women's futsal performance, which can help coaches to improve their performance.**

**Abstract:** Futsal is classified as a high-intensity intermittent sport or repeated-sprint sport. Explosive and very fast movements are performed with short reaction time, interspersed with playing time of 3 to 6 min during the 40 min match, at intensities of 85–90% of maximum HR. Performance factors such as agility, sprint repetition capacity, aerobic endurance capacity, lower body power, and speed are associated with the game actions. These performance factors can be affected by the athlete's body composition. The aim is to determine the relationship between the different physical and physiological performance parameters and body composition in top-level women's futsal players. The subjects of the study were 12 elite female futsal players ($25.17 \pm 4.75$ years old) competing in the First Division Spanish League. An anthropometric assessment was conducted by an ISAK level III anthropometrist for three days during the competitive period. The sum of 4, 6 and 8 skinfolds and body composition were calculated with anthropometric data. Performance tests were conducted to evaluate agility, ability to repeat sprints, velocity and the explosive power of lower extremities in the playing court with specific warm up and previous explication during 2 days in the same week as the anthropometric tests. The tests used for that purpose were: *t*-test, Yo-Yo test, repeat-sprint ability (RSA), speed test, and jump test (JS, CMJ and ABK). Pearson correlations were used to establish the different associations with a *p*-value < 0.05. The results showed a negative correlation between agility and the fat component, and a positive correlation between the muscle component and aerobic capacity, agility, speed, and ABK jump. Body composition plays a fundamental role in the development of performance-related skills in women's futsal.

**Keywords:** futsal; female athlete; performance; body composition; anthropometry; agility; speed; VO$_2$max; sprint ability

## 1. Introduction

The duration of a match can exceed the theoretical 40 min of the match by up to 80% due to stoppages of time when play is stopped for fouls, throw-ins or corners [1–3]. This aspect, together with the small size of the field of play ($40 \times 20$ m) and the unlimited number of substitutions, makes futsal a form of high-intensity intermittent sport (HIIS) or sport of repeated sprints [4]. Players perform mostly interval–fractional-type efforts, of submaximal and maximal intensity, and interspersed with active and incomplete recovery breaks (low-intensity activities or game breaks) [5]. Between substitutions of the same player, the playing time usually lasts between 3 and 6 min [3,6]. During the match, intensities of around 85–90%

of the HRmax are recorded, with this intensity maintained for 80% of the match time, along with a maximum oxygen consumption level of 75% [6,7]. In addition, an average of nine different actions per minute are performed, at least two of them being of high-intensity [8], and with these situations related to aerobic power [9]. Therefore, these intensities are difficult to maintain due to the accumulation of metabolites, such as the infold in blood lactate, the depletion of muscle phosphocreatine deposits, or changes in intermuscular coordination that affect contraction [10]. In light of these demands, and the fast and explosive game actions, several authors state that the repeat sprint ability (RSA), defined as the ability to repeat these high-intensity actions [11–15], should be considered when evaluating performance in indoor football, as it can be used to discriminate the competitive level of an indoor football player [16,17]. Likewise, the speed of the player, which also influences the speed of the actions and the game, is another determining factor [18]. Another performance factor to consider is agility [19], defined as a rapid movement of the whole body with changes in speed and direction in response to various stimuli [20], which can be decisive in neutralizing the actions of the opponent, reducing the risk of injury, and improving the player's performance [21]. On the other hand, in terms of technique, the explosive power of the lower extremities is decisive for ball striking and explosive actions [22]. Lastly, due to the rules of the game of futsal, players require high aerobic capacities, as they can cover up to 4.5 km per match, as well as a good development of anaerobic energy pathways [16].

All these sport performance parameters can be modified by the athlete's body composition [23]. The fat percentage can influence the sprint time and the body power assessed by the vertical jump. Other studies have seen a correlation between the percentage of muscle mass on agility or aerobic capacity in sports with similar characteristics to futsal, such as handball, hockey or volleyball [24–26]. There is scarce scientific evidence of anthropometric characteristics of female futsal players in relation to their competitive performance [27,28]. However, Kooshaki et al. (2014) associated the anthropometric characteristics of female players to key futsal skills (ball control, ball sprint speed, passing, dribbling, and shooting), concluding that there was a direct negative association between fat percentage and the ability to dribble and control the ball at maximum speed, being worse the higher the fat percentage [29].

Given the above, it can therefore be deduced that when comparing two teams with the same or similar technical–tactical ability, the one with a higher level of physical condition will have a greater chance of success [30]. The aim of the study was to evaluate body composition and performance characteristics in elite female futsal players and analyze associations between the measured characteristics.

## 2. Materials and Methods

### 2.1. Subjects

The study population was selected by non-probability, non-injury convenience sampling. The team consisted of 14 players, but two of them were excluded due to injury. The 12 Spanish elite female futsal players from the Alicante team ($25.17 \pm 4.75$ years old) voluntarily participated in this study. None of the players suffered any injuries in the 6 months prior to the tests on orthopedic or neurological diseases or operations of lower extremities. The team competes in the first division of the Spanish futsal league. Team's track record includes twice European University champions, qualifying for the Spanish queen's cup 9 times, 8 Spanish university championships, and runners-up in the Spanish league 2016/2017 season. The composition by positions of the sample was as follows: 3 goalkeepers, 4 forwards, 4 wingers and 1 pivot. The female futsal players studied had, at least, 5 years competing and training in elite futsal teams. The players trained a total of 7.5–12.5 h/week, spread over a total of 3–5 sessions depending on the period of the season they were in. The athletes took part in afternoon training sessions from Monday to Friday (strength, technical–tactical, endurance, and plyometrics) at 19:30. On weekends, they generally competed on Saturday evenings at 19:00 for home matches. For away matches every fortnight, the team travelled by bus and the journeys lasted between 4 and 26 h (round trip).

Anthropometric and performance assessment was performed in February 2020 during the competition period. The anthropometric evaluation was carried out in non-working or non-teaching hours in the afternoon, the previous training session, on 3 consecutive days. On the measurement days, the players should not have performed high-intensity exercise the previous day, nor should they have performed training or stretching sessions on the same day. Anthropometric assessment was performed in a room that was especially designed for this purpose [31,32]. The performance tests were carried out on the usual training track and at the usual training time, in the afternoon before the start of training in two sessions in the same week before the week of the anthropometric assessment. A written informed consent form was obtained from the participants explaining the objectives and characteristics of the study.

### 2.2. Experimental Design

This is an observational and descriptive study. The subjects studied included the elite female futsal players of the University of Alicante. Exclusion criteria were players with any active injuries.

### 2.3. Instrumental Procedures

2.3.1. Anthropometric Measurements

Anthropometric measurements were taken according to the ISO 7250-1:2017 and the International Society for the Advancement of Kinanthropometry (ISAK) standard [31].

The anthropometric measurements were performed using a measuring rod (1 mm accuracy), a scale (TANITA—BC-601, Amsterdam, The Netherlands—100 g accuracy), a flexible tape measure (1 mm accuracy), a small sliding caliper (Cescorf, Porto Alegre, Brasil—1 mm accuracy), a skinfold caliper (Holtain, Crymych, UK—0.2 mm accuracy) and others Supplementary Material (demographic pencil and anthropometric box).

A trained anthropometrist with level III ISAK certification conducted the assessment. Body composition measurements included (1) Basic measurements (body mass and stretch stature); (2) Skinfolds (triceps, subscapular, biceps, iliac crest, supraspinal, abdominal, thigh, and calf); (3) Girths (arm relaxed, arm flexed and tensed, waist, abdominal, hips, thigh middle, and calf); (4) Breadths (humerus, bi-styloid, and femur). The body composition was determined using the formulas described by the Spanish Kinanthropometry Group (GREC) [32], following the four-component model (muscle mass (MM), fat mass (FM), bone mass (BM) and residual mass (RM)). For this, the following were utilized: Wither's formula for calculating FM expressed in percentage [31], Lee's formula for calculating MM expressed in kg [33], and Rocha's formula for calculating BM expressed in kg [34]. At the same time, the sum of 8 skinfolds was calculated, as well as two health indices: the waist-height index and the body adiposity index. The technical errors of measurement were 0.04% for basic measurements, 2.34% for skinfolds, 0.26% for girths and 0.35% for breadths.

Somatotype was estimated following the Heath–Carter method, establishing the three Carter components (endomorph, mesomorph, and ectomorph, separately) and representing those results in a somatotype chart. The somatotype chart is the graphical representation of the somatotype where the rating of the three components of the somatotype is plotted in a two-dimensional chart [35].

2.3.2. Performance Testing

All tests were carried out during the second week of February 2020, coinciding with the second phase of the competition period, and after two weeks of readjustment to training. These were carried out during two training sessions, with a minimum of 48 h between sessions on the training track with the usual training clothes. In the first session, the jumping test, running speed test and RSA test were performed in that order. The agility test (*t*-Test) and aerobic endurance capacity test ("Yo-Yo") were performed in the second session in that order. A generic, general warm-up was carried out before participation in the tests, which consisted of 3 min of continuous running, 5 min of dynamic joint mobility, and

3 progressive sprints of 25 m. A rest period of 10 min was allowed between the different tests. All subjects were familiarized with the tests and had performed them at least once before, as well as with the explanations given by the researchers in situ.

Aerobic Endurance Capacity

A PolarM400® heart rate monitor with a PolarH7® chest strap was used to measure heart rate (HR). The "Yo-Yo Intermittent Recovery Test level 1" (YYIR1) was performed. It is an intermittent aerobic endurance test with a progressive infold in intensity until exhaustion. The test consists of periods of 40 m sprints ($2 \times 20$ m out-and-back), increasing in speed according to the distance covered, with the time being monitored by beeps from an audio device connected to a loudspeaker [36]. An active recovery time of 10 s is provided, which remains unchanged throughout the test, consisting of $2 \times 2.5$ m of jogging. The test is terminated when a subject fails to reach the finish line twice in succession. The infold in speed is carried out as follows: 4 runs at 10–13 km/h (0–160 m) and another 7 runs at 13.5–14 km/h (160–440 m), continuing with speed increments of 0.5 km/h, staggered after every 8 runs (i.e., after 760, 1080, 1400, 1720 m, etc.) until exhaustion. The track is bounded by cones, with a width of 2 m and a length of 20 m, with another cone marking the active recovery distance (5 m) [37]. The total distance covered in the test was used as a criterion for the estimation of $VO_2max$ [36]. The characteristics measured were the distance covered in meters and HR in beats per minute, giving rise to the calculation of $VO_2max$. The average values of maximum HR, average distance covered and average $VO_2max$ of all the players were used for the analysis.

RSA

Time was measured for 5 and 25 m using three photocells (Microgate® Polifemo Radio Light, Bolzano, Italy) (1 m from the floor). In the second session, an RSA test was performed to estimate the ability to repeat high-intensity actions, where each player had to perform 6 maximum sprints of 25 m with 25 s of active recovery between them [38], also measuring the acceleration in 5 m [17]. Recovery was performed by jogging back to the starting point, providing auditory feedback to inform when to start the sprint. The start mark was placed 0.5 m from the first photocell. As performance indicators, the sum of the times recorded to cover the 6 repetitions of 25 m [39], the fatigue index, associated with the ability to repeat sprints, using the sprint decline rate (Sdec), determined using the equation: Sdec = (RSAtotal/(RSA best $\times$ 6) $\times$ 100) $-$ 100 [14], and the change in fatigue index relating the first and last sprint using the equation: Change = ((RSA latest $-$ RSA first)/(RSA first)) $\times$ 100 [39].

Running Speed

Three photoelectric cells (Microgate Polifemo Radio Light, Bolzano, Italy) [17] were used as the instruments (1 m from the floor), which were placed at the start, at 5 m, to measure the acceleration time of the players, and at 25 m, to evaluate the total time at the end of the run. It should be noted that the players had the starting line 0.5 m from the first photocell. The average time it took the players to cover 5 and 25 m course were calculated, as well the best mark of both distances was recorded in seconds.

Agility

One photoelectric cell (Microgate Polifemo Radio Light, Bolzano, Italy) [17] was used as the instrument placed at the start of the drawn T. The *t*-Test was used, following the protocol with certain modifications made by Asencio-Vicedo et al. (2020) [40]. This test consists of a circuit at maximum speed in which a letter T has to be drawn and delimited with cones in 4 points (Figure 1). The player started the test with both feet 0.5 m behind the starting line (A) and sprinted in a straight line 10 m to point B. Once the cone was reached, the player shuffled to the left and 5 m to the next cone (C), after which the player shuffled to right 10 m to touch the cone on the other side (D). Then, shuffled to the left

5 m back to return to the central cone (B), in that moment, players ran backward until they passed the starting line (A). The *t*-test measures different types of displacement. Lateral and backwards. In futsal, there are not only linear movements, but also lateral movements (covers, and supports) and backwards movements (retreats and pivot play). Each subject was allowed a minimum of 3 min between two attempts to ensure adequate recovery. The measure obtained to establish correlations is the best time in seconds of three attempts.

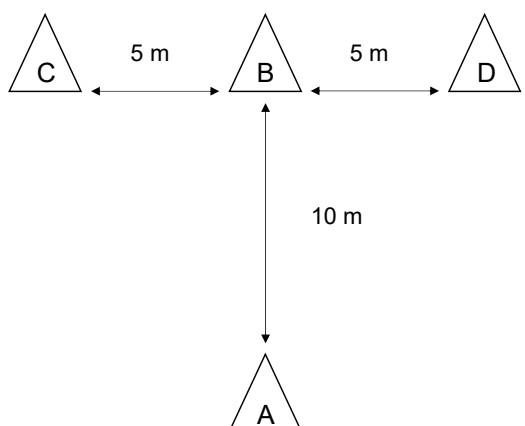

**Figure 1.** Graphical representation of the agility test "*t*-test".

Explosive Power of Lower Extremities

The Chronojump DIN-A4 force platform (Optojump, Microgate, Bolzano, Italy) was used together with the Chronojump software application. The procedure followed was a battery of the jump tests Squat Jump (SJ), Countermovement Jump (CMJ), Abalakov jump (ABK) [41] which follows the following sequence: Initially, the SJ is performed, which consists of a jump starting from a knee flexion of approximately 90°. The hands must be placed on the hips and must not be separated until the end of the jump. This jump is performed without a countermovement. Next, a CMJ is performed, which consists of starting the jump in the same way as the previous procedure, but this time, when starting the jump, the athlete can perform a countermovement or rebound, that is, perform a knee flexion–extension and then look for a vertical jump. Finally, an ABK is performed, which consists of a similar procedure to the previous ones, except that in this type of jump, the players can use the swinging of their arms to propel their bodies towards the maximum vertical jump. Up to 3 attempts were allowed for each jump with a recovery time between attempts of minimum 3 min to ensure complete recovery. The measurements recorded in these tests are the flight time in seconds and the height the players reach in cm. The best jump was used for the analysis.

*2.4. Statistical Analysis*

The Statistical Package for Social Sciences (SPSS) v.22 software was used to analyze the data. A Kolmogorov–Smirnov normality test indicated a normal distribution, so the statistical test applied was Pearson's correlation coefficient (r) to determine the relationships with the performance of all measured parameters. Variables with a correlation between 0.30 and 0.70 were considered as moderate associations, while variables with a correlation higher than 0.70 were considered as sufficiently strong associations. *p*-values lower than 0.05 were considered statistically significant [42].

**3. Results**

Tables 1 and 2 shows the anthropometric values of the sample derived from the anthropometric measurements taken.

**Table 1.** Anthropometric data.

| Variable | Mean | SD |
|---|---|---|
| Basic measures | | |
| Body mass (kg) | 59.79 | 6.36 |
| Stretch stature (m) | 1.64 | 0.05 |
| BMI (kg/m$^2$) | 22.27 | 1.75 |
| **Skinfolds** | | |
| Subscapularis (mm) | 9.41 | 2.67 |
| Triceps (mm) | 13.76 | 3.43 |
| Biceps (mm) | 4.83 | 1.66 |
| Iliac crest (mm) | 13.73 | 5.02 |
| Supraspinal (mm) | 9.33 | 2.64 |
| Abdominal (mm) | 14.69 | 5.34 |
| Thigh (mm) | 22.93 | 4.68 |
| Calf (mm) | 10.20 | 2.72 |
| Sum of 4 skinfolds (mm) | 42.70 | 8.64 |
| Sum of 6 skinfolds (mm) | 80.32 | 15.84 |
| Sum of 8 skinfolds (mm) | 101.89 | 21.78 |
| **Girths** | | |
| Relax arm (cm) | 26.34 | 1.84 |
| Flexed and tensed arm (cm) | 27.38 | 1.64 |
| Waist (cm) | 70.31 | 3.31 |
| Abdominal (cm) | 76.81 | 4.64 |
| Hips (cm) | 96.29 | 3.93 |
| Thigh middle (cm) | 52.01 | 2.34 |
| Calf (cm) | 34.98 | 1.52 |
| **Breadths** | | |
| Humerus | 6.11 | 0.33 |
| Bi-styloid | 5.01 | 0.24 |
| Femur | 9.14 | 0.32 |

**Table 2.** Body composition data.

| Variable | Mean | SD |
|---|---|---|
| Body Density | 1.07 | 0.01 |
| Fat Mass (FM) (%) | 21.03 | 3.05 |
| Fat Mass (FM) (Kg) | 12.57 | 3.29 |
| Muscle Mass (MM) (%) | 38.39 | 2.05 |
| Muscle Mass (MM) (Kg) | 22.95 | 5.80 |
| Bone mass (%) | 16.00 | 1.19 |
| Bone mass (Kg) | 9.57 | 2.47 |
| Endomorphy | 3.43 | 0.79 |
| Mesomorphy | 3.53 | 0.68 |
| Ectomorphy | 2.13 | 0.88 |

FM: Wither's formula; MM: Lee's formula; BM: Rocha's formula. Whiter's formula: $100 \times (4.95/\text{Body Density} - 4.5)$; Lee's Formula: Height $\times$ ($0.00744 \times$ perimeter corrected arm contracted$^2$ + $0.00088 \times$ corrected front thigh girth$^2$ + $0.00441 \times$ calf girth$^2$) + ($2.4 \times$ age) − ($0.048 \times$ gender) + race + 7.8; Rocha's formula: $3.02 \times$ (Height$^2$ $\times$ Bystiloid breadth/100 $\times$ Femur breadth/100 $\times$ 400)$^{0.712}$.

Table 3 shows the results of the different performance tests carried out, expressed in the measurement value that corresponds to each of them as well as the standard deviation (SD) of each value.

**Table 3.** Performance testing results.

| Performance Test | Value | SD |
|---|---|---|
| ***t*-test** | | |
| Best (s) | 11.01 | 0.31 |
| **Running Speed** | | |
| Mean 5 m (s) | 1.16 | 0.08 |
| Best 5 m (s) | 1.14 | 0.08 |
| Mean 25 m (s) | 4.07 | 0.15 |
| Best 25 m (s) | 4.03 | 0.16 |
| **CMJ, SJ and ABK** | | |
| Best Flight time SJ (s) | 0.49 | 0.02 |
| Best Height SJ (cm) | 28.95 | 2.57 |
| Best Flight time CMJ (s) | 0.50 | 0.02 |
| Best Height CMJ (cm) | 31.47 | 2.82 |
| Best Flight time ABK (s) | 0.52 | 0.03 |
| Best Height ABK (cm) | 33.60 | 3.75 |
| **Yo-Yo** | | |
| HR Max (bpm) | 189.78 | 2.05 |
| Average HR (bpm) | 167.22 | 10.23 |
| Distance (m) | 1120.00 | 336.4 |
| $VO_2$max (mL/kg/min) | 45.81 | 2.83 |
| **RSA** | | |
| Mean (s) | 3.87 | 0.16 |
| Total (s) | 23.30 | 0.98 |
| Best (s) | 3.76 | 0.18 |
| Sdec (s) | 2.79 | 1.62 |

HR Max: Heart Rate Maximum. Maximum number of beats per minute that the heart reached during the test; Sdec: Sprint Decline Rate, bpm: beats per minute.

The correlations between the values in Tables 1 and 2 are shown in Table 3, where the main variable, the test in question, and the anthropometric factor to which it is related are shown in Table 4.

The Yo-Yo aerobic endurance test showed a medium–high inverse correlation between $VO_2$max values with abdominal skinfold (r = −0.772) and abdominal girth (r = −0.685). In addition, a correlation with the mesomorphic component (r = 0.672) was also found.

In turn, agility also showed a strong correlation with the abdominal skinfold (r = 0.744), and moderately high correlations with the biceps fold (r = 0.610), endomorphy (r = 0.616) and the different sums of skin folds, as shown by r = 0.575 (sum of 4 folds), r = 0.593 (sum of six skinfolds) and r = 0.576 (sum of eight skinfolds). There were no relevant correlations between agility and other variables.

As for jumping, there was a moderately high correlation between the size of the players and the flight time in the ABK jump (r = 0.605) and with the height of the same type of jump (r = 0.598). In relation to the other types of jumps that were measured, no relevant correlations were observed, neither with the SJ nor with the CMJ.

Correlations were also found between speed and the mesomorphic component, as observed by a moderately low correlation with the 5 m mean (r = 0.430), and moderately high correlation with the 25 m mean (r = 0.612). Finally, a moderately high correlation was found between mesomorphy and the average of each player's best 25 m performance (r = 0.624).

The RSA test did not present any significant correlation that could relate any anthropometric parameter to the RSA capacity.

**Table 4.** Correlations between test and anthropometric variables.

| Test | Variable Performance | Variable Anthropometric | Pearson's Correlation | *p*-Value |
|---|---|---|---|---|
| *t*-test | Agility | Biceps skinfold | 0.610 | 0.035 * |
| | | abd. skinfold | 0.744 | 0.006 * |
| | | Sum. 4 skinfolds | 0.575 | 0.050 * |
| | | Sum. 6 skinfolds | 0.593 | 0.042 * |
| | | Sum. 8 skinfolds | 0.576 | 0.050 * |
| | | Endomorphy | 0.616 | 0.033 * |
| Flight time ABK | Explosive power | Stretch stature | 0.605 | 0.037 * |
| Height ABK | | Stretch stature | 0.598 | 0.040 * |
| Mean 5 m. | Speed | Mesomorphy | 0.430 | 0.163 |
| Best 5 m. | | Mesomorphy | 0.512 | 0.089 |
| | | Size | 0.472 | 0.122 |
| | | Sum. 4 folds | 0.419 | 0.175 |
| Mean 25 m. | | Mesomorphy | 0.612 | 0.034 * |
| | | Ectomorphy | 0.478 | 0.116 |
| | | % fat | 0.435 | 0.158 |
| Best 25 m. | | Mesomorphy | 0.624 | 0.030 * |
| Yo-Yo | VO$_{2max}$ | Ectomorphy | 0.458 | 0.134 |
| | | abd. skinfold | −0.772 | 0.015 * |
| | | abd. girth | −0.685 | 0.042 * |
| | | Biceps skinfold | −0.537 | 0.136 |
| | | Iliac crest fold | −0.633 | 0.067 |
| | | Suprasp. skinfold | −0.516 | 0.155 |
| | | Sum. 8 skinfolds | −0.541 | 0.133 |
| | | BMI | −0.557 | 0.119 |

* *p*-value: Significant *p* value ($p < 0.05$). abd: abdominal; Suprasp: supraspinal, sum: summation.

## 4. Discussion

The aim of this study was to evaluate body composition and performance characteristics in elite female futsal players and analyze associations between the measured characteristics. This study showed that different characteristics of body composition and anthropometric measures presented correlations with performance parameters. Aerobic endurance capacity was inversely correlated with abdominal skinfold and abdominal girth and positively correlated with the mesomorphic component. The abdominal skinfold also was related positively with agility, that could be affected by the endomorphy component and the sum of four and four skinfolds. The highest players obtained better results in the ABK jump and players who had more mesomorphy component obtained better results in the 25 m velocity test. The RSA test was the only test with no relation with any performance parameter.

In other studies, body composition and anthropometric variables have shown some relationship with certain components of athletic performance [43–45]. Body fat percentage is one of the limiting factors in terms of the ability to perform explosive actions [46]. The results obtained in this study in terms of fat mass percentages (21.30 ± 3.51) were similar to the percentages found in the scientific literature with athletes in sports with the same characteristics [27,28,47–49]. On the other hand, the mean percentage of muscle mass obtained (38.39 ± 2.05) was below that obtained by Queiroga et al., 2018 [49] in a team of female indoor football players, where they presented percentage levels of 45.4 ± 4.5. However, the result concurred with the study carried out by Voser (2016) [49], whose mean lean mass percentage was 38.32 ± 1.6. Regarding somatotype, the players showed an endo-mesomorphic body structure (3.43–3.53–2.13), which means that there is a balance between fat percentage and muscle percentage. Similar values were presented in a study carried out on 115 female players in Brazil (4.5–4.1–2.0), with higher values than those obtained in our

study, but with the same body classification [50], just as in another study carried out with 86 Spanish players with similar values and the same classification (3.82–3.27–2.01) [48].

These body composition variables could be related to a greater or lesser extent with some skills necessary for sports practice, such as agility, aerobic capacity, lower body power, and speed. In relation to agility, the time obtained in the *t*-test measurement was higher than those obtained in other studies with athletes trained in other sports with similar characteristics [40]. In contrast, the scientific literature has described a direct relationship between the assessment of aerobic capacity in a level 1 intermittent endurance Yo-Yo test, and the physical performance in an indoor football match, of high-level competitive players [51]. The mean $VO_2max$ values obtained in this study were within the mean values found in studies related to intermittent sports [51–53] and there was no clear evidence that higher $VO_2max$ values significantly improved competitive performance in this discipline [53]. Another of the determining variables of performance in indoor football is the explosive power of lower extremities [22]. In the study carried out by Gorostiaga et al. (2009) [54], it was observed that indoor football players obtained lower values in the vertical jump (38.1 cm) than outdoor football players (44.9 cm), which may be due to the fact that in indoor football, there are fewer frequent actions in which vertical jumps are performed or needed, as the ball travels for a longer period at ground level. The results of our study (31.47 cm) and another study carried out with elite female futsal players (26.7 cm) [9] showed lower values than those obtained in the male sex due to the greater muscle power developed in the male sex. As for the CMJ and SJ vertical jumps, no correlations were found with the anthropometric variables. On the other hand, the flight time and height of the ABK jump correlated with the height of the female players. The highest correlations obtained in our study are shown between mesomorphy and the time employed running 25 m, both in the overall mean (r = 0.612) and the mean of the best times (r = 0.624). To measure speed in futsal, some studies have used different distances [38,55], and 25 m may be a slightly excessive distance in futsal [56]. After analyzing the results of our study, and in comparison with the literature found, we can say that a percentage of muscle mass, related to the mesomorphic component, could improve the speed factor, as documented in other studies related to acceleration and speed and muscle mass [57–59], with this being an advantage for indoor football players when performing technical and motor actions during a match [9].

From the results of our study, a mean inverse correlation between fat percentage and total $VO_2max$ derived from the Yo-Yo test was found (r = −0.590; *p* = 0.094); this was also corroborated by the data obtained from the skinfolds, which also showed a significant inverse correlation for the parameters of abdominal skinfold with $VO_2max$ (r = −0.772; *p* = 0.015) and abdominal perimeter with $VO_2max$ (r = −0.685; *p* = 0.042). Furthermore, a correlation was also observed between muscle mass values and aerobic capacity (r = 0.513; *p* = 0.158), which may indicate that levels of muscle mass close to 40%, and low levels of fat mass, could contribute to the development of good aerobic capacity. Our study showed a strong correlation between agility and anthropometric variables related to the fatty compartment (sum skinfold, abdominal skinfold, biceps skinfold, and endomorphic component), indicating that the lower the value measured in these variables, the greater the agility that can be developed for the performance of actions typical of indoor football such as changes in direction, accelerations, or decelerations, but no references have been found in the scientific literature that relate agility to these variables. Finally, in our study, no correlations were found between the ability to repeat sprints and anthropometric variables, but other studies have associated a greater ability to repeat sprints with a high muscle profile and low adiposity [60], which may be due to the sample size and gender of the subjects.

The limitations of our study are centered on the lack of other studies that focused on female futsal players and studies of futsal and similar sports that relate body composition to physical performance. On the other hand, the authors consider the sample size to be a limitation; it should be taken into consideration that there was only one first division women's futsal team in the province of Alicante, and due to the logistics of the tests, it was impossible to increase the sample with teams from other autonomous communities. It is

also worth mentioning the timing of the anthropometric measurements. In our study, these measurements were taken in the afternoon because the players are students and many of them work, which is why in the mornings, they could not go to the room prepared for the tests, having to transfer the analyses to the afternoons, before the training session. In this sense, and due to the small sample size, the body composition and performance tests were not distinguished by the different positions of the players, as the sample would be very small, and it would not be possible to establish significant relationships between the different groups by position. This aspect should be considered for future research, as the difference between positions may exist and define the performance characteristics of a player.

## 5. Conclusions

Aerobic capacity, $VO_2$max and speed are related to a predominantly mesomorphic component, higher total muscle mass and lower % and total fat mass % of each player. A larger abdominal skinfold and a larger abdominal girth are associated with poorer agility and the stretch stature of the players is related to greater lower-extremity explosive power. The development of a balanced somatotype between both, with a slight mesomorphic predominance, seems to offer a better performance in women's indoor football. Key factors in futsal performance, such as aerobic capacity, $VO_2$max and agility, benefit when athletes have anthropometric parameters with a higher percentage of muscle mass. The results of this study may be useful for the coaching staff of women's futsal teams, who can assess the anthropometric composition of the players and thus determine what could be changed to improve performance. It also generates data for comparison in future research in the field of women's futsal related to body composition and performance.

**Author Contributions:** Conceptualization, J.M.M.-S., I.S., J.A.H.-S. and M.C.; methodology, J.M.M.-S., I.S., E.G.-R., S.S. and A.P.-T.; data curation, M.C., A.P.-T. and S.S. formal analysis, A.P.-T., J.A.H.-S., S.S. and E.G.-R.; investigation, M.C., A.P.-T., S.S. and J.M.M.-S.; visualization, J.M.M.-S. and I.S.; writing—original draft preparation M.C., A.P.-T., J.A.H.-S. and E.G.-R.; writing—Review and editing, J.M.M.-S., I.S., A.P.-T., E.G.-R., J.A.H.-S., S.S. and M.C. All authors have read and agreed to the published version of the manuscript.

**Funding:** This research received no external funding.

**Institutional Review Board Statement:** The study was conducted in accordance with the Declaration of Helsinki and approved by Ethics Committee of University of Alicante (protocol code UA-2018-05-22).

**Informed Consent Statement:** Written informed consent has been obtained from the participants to publish this paper.

**Data Availability Statement:** The data presented in this study are available on request from the corresponding author. The data are not publicly available due to privacy reasons.

**Acknowledgments:** The authors would like to thank all the research subjects for their willingness and interest in participating, as well as the technical staff of the indoor football team and the sports service of the University of Alicante for their predisposition in the organization and development of all the tests. We are grateful to Mario Fon (native English speaker) for their review of the English grammar and style of the current report.

**Conflicts of Interest:** The authors declare no conflict of interest.

## Abbreviations

Muscle mass (MM), fat mass (FM), residual mass (RM), bone mass (BM), heart rate (HR), countermovement jump (CMJ), Abalakov jump (ABK), squat jump (SJ), maximal oxygen consumption ($VO_2$max), repeated-sprint ability (RSA), Whiter's formula: $100 \times (4.95/\text{Bodyl Density} - 4.5)$; Lee's formula: Height $\times (0.00744 \times \text{perimeter corrected arm contracted}^2 + 0.00088 \times \text{corrected front thigh girth}^2 + 0.00441 \times \text{calf girth}^2) + (2.4 \times \text{age}) - (0.048 \times \text{gender}) + \text{race} + 7.8$; Rocha's formula: $3.02 \times (\text{Height}^2 \times \text{Bystiloid breadth}/100 \times \text{Femur breadth}/100 \times 400)^{0.712}$, Heart Rate Maximum (HR Max); Sprint Decline Rate (Sdec), beats per minute (bpm); Abdominal (abd); supraspinal (Suprasp).

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
