# Peer review of "Relationship between Body Composition and Performance Profile Characteristics in Female Futsal Players"

_applsci, doi:10.3390/app122211492_

Round 1

Reviewer 1 Report

Journal                      Applied Science

Manuscript                applsci-1947756-peer-review

Title of paper           

Relationship between body composition and performance profile in female futsal players

General comments

The main aim of the study was to analyze associations between body composition and performance characteristics in female futsal players. Participants were 12 elite female futsal players aged 25.2 (SD 4.8) years. Anthropometric characteristics were measured, and body composition parameters were calculated by formulas. T-test, Yo-Yo test, repeat sprint ability, running (speed) test of 5 and 25 m, jump test have been performed during competition period. Correlation analysis of the measured characteristics has been analyzed. 

The aim, methods and results have been described with sufficient clarity.  The references give an overview of the studies in the topic of research. A positive aspect of the present study is that researchers measured anthropometric characteristics according to ISAK rules.  Data is presented in three tables. In the Discussion chapter the data collected by the authors are compared with other research reports, and conclusions include some practical recommendations for this kind of sport.

Specific comments

It is recommended to add the word “characteristics” in the title of the manuscript, e.g. “Relationship between body composition and performance profile characteristics in female futsal players”.

The key words should be in the same sequence in the abstract as they occur in the description of the main findings and in conclusions.

Abstract

Modify the abstract´s content: aim, participants, add the description of methods (1. Anthropometric characteristics, 2. Performance tests (concrete), results, conclusions.

L17: please replace “little” with “short” [reaction time].

Introduction

Please merge one-two-sentence paragraphs into one.

L68 please correct the beginning of the sentence “Little scientific evidence…”

L79 please change the sentence “Due to the lack..  is to determine  whether there is any relationship…” to “The aim of the study was to evaluate body composition and performance characteristics in elite female futsal players and analyze associations between the measured characteristics”.

Materials and Methods

Participants

L86 please use for the age data one place after comma. Is ± SE or SD used here?

Please add exclusion criteria here if they were applied. Did the participants have any previous injuries, orthopaedic or neurological diseases/operations of lower extremity?

Were participants familiarized with the study protocol and performing of exercises?

Anthropometric measurement

L112. Abbreviation BC for body composition has not been used in the text, please delete it.

Please add formulas in the abbreviations list.

L123. Measurement errors – were they calculated by authors or accepted from reference (in this case please add the reference).

Performance testing

Please begin all descriptions of tests from the characteristics of applied devices.

L145-146 Please change unit of speed “km-h” to “km/h”

L150 Please add which characteristics of Yo-Yo test were measured. Please add description of FC max characteristic (Table 1).

L166 Please replace “Speed” with “Running speed”

L165. Please add which characteristics of running speed were measured. Average or mean, 5 and 25 m run? The same description has to be included in Table 1.

L171 Please add which agility characteristics were measured? What is media (s) (Table 1)?

L178 Please replace “lower body power” with “Explosive power of lower extremities”.

L182 Bosco test consists in repeated jumps during 30 or 60 seconds. In this study three kind of jumps have been performed – squat jump, counter-movement jump, counter-movement jump with the help of arms.

L193 Please add which characteristics were taken for analysis – the best result (highest jump) or mean of 3 attempts?

Results

The measured characteristics should be presented similarly in the text of Methods and in all tables. Each table should include in the notes the used abbreviations.

Table 1.

Please split data of Table 1 into two tables:

1.      Skinfold data

2.      Body composition data – weight, height (is it meant by the word “size (m)”?), BMI, etc.

What is DC? A unit?

Please use the word “height” instead of “size”.

Table 2.

What is DE? Is it SD?

Please add explanation for tests:

Agility: T-test performance time, s (media, s) ?

Running speed: 5m (mean, best), 25 m (mean, best)

Explosive power of lower extremities: SJ height (time, s; height, cm); CMJ (time, s; height, cm); CMJ with the help of arms ((time, s; height, cm).

Aerobic endurance capacity: …

Repeat sprint ability: Time of 6 x25m + 6x5m (s) ;

What is “Media?”

Spring decline rate or sprint decay index (see in L162)?

Table 3

Please replace “Lower train  power” with “Explosive power”

What is size? Please change this word for “height”.

What is media? Running speed?

Please check all terms in the left column.

Mesomorphy and “best  25 m” correlation p-value *p<0.05.

Discussion

Please begin the first paragraph with the description of the main results of the present study.

Conclusions

Please add concrete conclusions based on the main results of the present study (correlations); and add some remarks for practical use and future studies.

 Abbreviations: Please check all abbreviations used in the article.

Author Response

The main aim of the study was to analyze associations between body composition and performance characteristics in female futsal players. Participants were 12 elite female futsal players aged 25.2 (SD 4.8) years. Anthropometric characteristics were measured, and body composition parameters were calculated by formulas. T-test, Yo-Yo test, repeat sprint ability, running (speed) test of 5 and 25 m, jump test has been performed during competition period. Correlation analysis of the measured characteristics has been analyzed. 

The aim, methods and results have been described with sufficient clarity.  The references give an overview of the studies in the topic of research. A positive aspect of the present study is that researchers measured anthropometric characteristics according to ISAK rules.  Data is presented in three tables. In the Discussion chapter the data collected by the authors are compared with other research reports, and conclusions include some practical recommendations for this kind of sport.

Specific comments

It is recommended to add the word “characteristics” in the title of the manuscript, e.g. “Relationship between body composition and performance profile characteristics in female futsal players”.

Response of the authors: According to the reviewer's suggestions, the title has been modified.

The key words should be in the same sequence in the abstract as they occur in the description of the main findings and in conclusions.

Response of the authors: Following the reviewer's suggestions, the sequence of key words has been changed.

Abstract

Modify the abstract´s content: aim, participants, add the description of methods (1. Anthropometric characteristics, 2. Performance tests (concrete), results, conclusions.

Response of the authors: Following the reviewer's suggestions, changes have been done.

L17: please replace “little” with “short” [reaction time].

Response of the authors: the change has been done.

Introduction

Please merge one-two-sentence paragraphs into one.

Response of the authors: Following the reviewer's suggestions, changes has been done.

L68 please correct the beginning of the sentence “Little scientific evidence…”

Response of the authors: Following the reviewer's suggestions, the sentence has been changed.

L79 please change the sentence “Due to the lack..  is to determine whether there is any relationship…” to “The aim of the study was to evaluate body composition and performance characteristics in elite female futsal players and analyze associations between the measured characteristics”.

Response of the authors: Following the reviewer's suggestions, changes has been done.

Materials and Methods

Participants

L86 please use for the age data one place after comma. Is ± SE or SD used here?

Response of the authors: Following the reviewer's suggestions, changes has been done. We are using SD.

Please add exclusion criteria here if they were applied. Did the participants have any previous injuries, orthopedic or neurological diseases/operations of lower extremity?

Response of the authors: Thank you for your suggestion. The sentence “None of the players suffered any injuries in the 6 months prior to the tests orthopedic or neurological diseases or operations of lower extremities.” Has been added in this section.

Were participants familiarized with the study protocol and performing of exercises?

Response of the authors: Following the reviewer's suggestions, the authors added a sentence in order to explain this question in section “2.3.2 Performance testing”.

Anthropometric measurement

L112. Abbreviation BC for body composition has not been used in the text, please delete it.

Response of the authors: According to the reviewer's suggestions, the change has been done.

Please add formulas in the abbreviations list.

Response of the authors: Formulas have been added in abbreviation list.

L123. Measurement errors – were they calculated by authors or accepted from reference (in this case please add the reference).

Response of the authors: According to the reviewer's suggestions, the technical errors of measurement has been indicated for the anthropometric measurements made (0.04% for basic measurements, 2.34% for skinfolds, 0.26% for girths and 0.35% for breadths).

Performance testing

Please begin all descriptions of tests from the characteristics of applied devices.

Response of the authors: According to the reviewer's suggestions, the characteristics of applied devices have been added at the beginning of the tests’ description.

L145-146 Please change unit of speed “km-h” to “km/h”

Response of the authors: The change has been done.

L150 Please add which characteristics of Yo-Yo test were measured. Please add description of FC max characteristic (Table 1).

Response of the authors: Following the reviewer's suggestions, the characteristics measured for Yo-Yo test have been added and FCmax has been changed by HR (heart rate) and added the characteristics in table 1.

L166 Please replace “Speed” with “Running speed”

Response of the authors: Following the reviewer's suggestions, changes has been done.

L165. Please add which characteristics of running speed were measured. Average or mean, 5 and 25 m run? The same description has to be included in Table 1.

Response of the authors: The total time of the 25-metre sprint was measured, but a photocell was also placed at 5 meters for the acceleration speed from 0 to 5 meters, as this is a sport where there is a lot of constant acceleration and deceleration. As has been previously described, rarely does the player reach her maximum sprint speed in a match, because the average sprint per match is 10 meters (Spyrou K, Freitas TT, Marín-Cascales E, Alcaraz PE. Physical and Physiological Match-Play Demands and Player Characteristics in Futsal: A Systematic Review. Front Psychol. 2020 Nov 6;11:569897. doi: 10.3389/fpsyg.2020.569897. PMID: 33240157; PMCID: PMC7677190). This information has been added in the correspondent section.

L171 Please add which agility characteristics were measured? What is media (s) (Table 1?

Response of the authors: Following the reviewer's suggestions, the authors added agility characteristics recorded with T-test.

L178 Please replace “lower body power” with “Explosive power of lower extremities”.

Response of the authors: The change has been done.

L182 Bosco test consists in repeated jumps during 30 or 60 seconds. In this study three kind of jumps have been performed – squat jump, counter-movement jump, counter-movement jump with the help of arms.

Response of the authors: Thanks for the appreciation, we only used a battery of tests commonly used by other authors to assess explosive strength in the lower extremities.  In this case we measured SJ, CMJ and ABK jumps.

Ruscello B, Castagna C, Carbonaro R, Gabrielli PR, D'Ottavio S. Fitness profiles of elite male Italian teams handball players. J Sports Med Phys Fitness. 2021 May;61(5):656-665. doi: 10.23736/S0022-4707.21.11850-X. Epub 2021 Jan 22. PMID: 33480511. 

L193 Please add which characteristics were taken for analysis – the best result (highest jump) or mean of 3 attempts?

Response of the authors: The characteristic used to perform the analysis were the best jump values and this information has been added in the manuscript. This criterion has been adopted according to previous literature, as the following study, where the best jump is also used to perform the analysis:

Nakamura FY, Pereira LA, Moraes JE, Kobal R, Kitamura K, Cal Abad CC, Teixeira Vaz LM, Loturco I. Physical and physiological differences of backs and forwards from the Brazilian National rugby union team. J Sports Med Phys Fitness. 2017 Dec;57(12):1549-1556. doi: 10.23736/S0022-4707.16.06751-7. Epub 2016 Nov 4. PMID: 27813392.

 Results

The measured characteristics should be presented similarly in the text of Methods and in all tables. Each table should include in the notes the used abbreviations.

 Table 1.

Please split data of Table 1 into two tables:

  1. Skinfold data
  2. Body composition data – weight, height (is it meant by the word “size (m)”?), BMI, etc.

What is DC? A unit?

Please use the word “height” instead of “size”.

 Response of the authors: Following the reviewer's suggestions: All the measurements are presented similarly among all the manuscript.

Table 1 has been split on two and “size” has been replaced with “height”. The words DC has been replaced by the correct one, Body Density.

Table 2.

What is DE? Is it SD?

Response of the authors: The correct term is SD and has been corrected in the text.

Please add explanation for tests:

Agility: T-test performance time, s (media, s)?

Running speed: 5m (mean, best), 25 m (mean, best)

Explosive power of lower extremities: SJ height (time, s; height, cm); CMJ (time, s; height, cm); CMJ with the help of arms ((time, s; height, cm).

Aerobic endurance capacity: …

Repeat sprint ability: Time of 6 x25m + 6x5m (s);

Response of the authors: Following the reviewer’s suggestions, all the test in table 3 have been improved with more information.

What is “Media?”

 Response of the authors: It is a mistake, Media is Mean. It has been corrected.

Spring decline rate or sprint decay index (see in L162)?

Response of the authors: The correct is Sprint Decline Rate. It has been corrected.

Table 3

Please replace “Lower train power” with “Explosive power”

What is size? Please change this word for “height”.

What is media? Running speed?

Please check all terms in the left column.

Mesomorphy and “best 25 m” correlation p-value *p<0.05.

 Response of the authors: thanks for the suggestions, “lower train power” and “size” have been replaced and “media” has been changed by “mean”. Correlation between mesomorphy and “best 25m” has been added.

Discussion

Please begin the first paragraph with the description of the main results of the present study.

Response of the authors: Following the reviewer’s suggestions, the first paragraph of the discussion has been modified.

Conclusions

Please add concrete conclusions based on the main results of the present study (correlations); and add some remarks for practical use and future studies.

Response of the authors: The authors are grateful for the reviewer's comments. The authors added the recommended information to conclusions section.

 Abbreviations: Please check all abbreviations used in the article.

Response of the authors: According to reviewer's comments the abbreviations used have been checked.

Reviewer 2 Report

Thank you for your submission looking at body composition and performance in female futsal athletes. While it is an interesting topic, the paper needs significant improvements in writing and organization. The intro and discussion, in particular, must be improved before publication. Please see comments below:

Introduction

·      Delete P1. Instead, begin the paper discussing futsal physical demands – we know it’s a team sport and doesn’t serve as a purposeful introduction.

·      Combine P2 and P3 into one paragraph discussing the physical and physiological demands associated with futsal

·      RSA and agility paragraphs can be combined. This paragraph should be synthesized as…two important elements associated with the success of futsal included RSA and agility. And discuss each.

·      You only discuss RSA and agility as being important in futsal, but then your purpose statement also contains aerobic capacity and lower body power. May want to add 2-3 sentences highlighting the importance of aerobic fitness and power, as a result of the physical demands associated with futsal

Methods

·      Change population to “participants” or “subjects”

·      Remove “.” After 2.3.1 and 2.3.2 titles

·      Avoid using bullet points. These should be subheadings

·      Add reference for r-values

Results

·      Table 2: what is value and DE? Define abbreviations under your table. This table is confusing

Discussion

·      P1: this should be going over the main findings of the study What were your main results?

·      P2: this should be discussing the body comp data you found – relative to other research, maybe how it compares to other sports?

·      P3: results of performance testing – relative to other research and how it compares to other similar sports

·      P4: discuss major correlations. Why did your study find no correlations and others have? Discussion may be warranted

Author Response

Thank you for your submission looking at body composition and performance in female futsal athletes. While it is an interesting topic, the paper needs significant improvements in writing and organization. The intro and discussion, in particular, must be improved before publication. Please see comments below:

Introduction

  • Delete P1. Instead, begin the paper discussing futsal physical demands – we know it’s a team sport and doesn’t serve as a purposeful introduction.
  • Combine P2 and P3 into one paragraph discussing the physical and physiological demands associated with futsal
  • RSA and agility paragraphs can be combined. This paragraph should be synthesized as…two important elements associated with the success of futsal included RSA and agility. And discuss each.
  • You only discuss RSA and agility as being important in futsal, but then your purpose statement also contains aerobic capacity and lower body power. May want to add 2-3 sentences highlighting the importance of aerobic fitness and power, as a result of the physical demands associated with futsal.

Response of the authors: Following the reviewer's suggestions, changes have been done.

Methods

  • Change population to “participants” or “subjects”

Response of the authors: The change has been done.

  • Remove “.” After 2.3.1 and 2.3.2 titles

Response of the authors: The changes have been done.

  • Avoid using bullet points. These should be subheadings

Response of the authors: Following the reviewer's suggestions, changes have been done.

  • Add reference for r-values

Response of the authors: The reference has been added.

Results

  • Table 2: what is value and DE? Define abbreviations under your table. This table is confusing

Response of the authors: The authors changed DE by the correct term, SD. Abbreviations were defined below the table.

Discussion

  • P1: this should be going over the main findings of the study What were your main results?
  • P2: this should be discussing the body comp data you found – relative to other research, maybe how it compares to other sports?
  • P3: results of performance testing – relative to other research and how it compares to other similar sports
  • P4: discuss major correlations. Why did your study find no correlations and others have? Discussion may be warranted

Response of the authors: Discussion section has been rewritten to improve the manuscript according to reviewer's suggestions.

Reviewer 3 Report

Dear Authors,

This study is too descriptive with low power and a too small sample. Also, methodological errors are too big to be addressed at this stage.  Therefore I recommend rejection as this paper is not suitable for a high level impact factor journal.

Abstract needs conclusion and the second sentence should be deleted. Add info on how was the body morphology measured. 

Introduction

Too many paragraphs. Logically join them to 3 max.

Sentence lines 63-68 is too long and hard to understand. Rewrite

Line 68-69 / Please add all available research in this field and comment on them in the discussion. Add:

 https://www.researchgate.net/publication/354381111_The_Relationship_between_Anthropometric_Characteristics_and_Motoric_Performance_of_Female_Futsal_Players

Methods

How did you determine the sample size (G*Power or any other method)? Report

What is their training experience and how many times per week do they train? report

The sample size is very small and the overall power of your study is extremely low. This is perhaps a solid case study.

Line 96 - was obtained from participants

Line 101 - what about previous injuries in like 6 months before testing? This could also influence performance

Why were anthropometric measurements taken in the afternoon and not in the morning? 

The technical error of measurements was 7.5%. Also, how was this measured - you need to report this. Sorry but this is very high/too high for a level 3 Isak accredited anthropometrist? It should be max 5% or less. This means your data is not reliable, therefore all your conclusions are doubtful.

Line 166 - how many repetitions did they make?

The agility test was not done according to the reference used. 10 yards and 5 yards are not equal to 10m and 5m! Therefore you did not perform the same test! This is a scientific paper and not a magazine article so you should be specific! Also when they reach the B point - they don't turn left and run they shuffled to the left which is a different movement. How many times was this tested? Why did you use the stopwatch and not the micro gate system as you used in the RSA test?

Lower body power - what was the break between tests? which result was taken to further analysis?

What was the break between tests and exact order?

The limitations paragraph is modest and should address error rates, anthropometrical measurements were done in the afternoon and putting together all different playing positions.

On the basis of this, I am rejecting this paper. However, I hope the feedback will be helpful to authors in further studies and in corrections.

Kind regards

Author Response

Reviewer 3

This study is too descriptive with low power and a too small sample. Also, methodological errors are too big to be addressed at this stage.  Therefore I recommend rejection as this paper is not suitable for a high level impact factor journal.

Abstract needs conclusion and the second sentence should be deleted. Add info on how was the body morphology measured. 

Introduction

Too many paragraphs. Logically join them to 3 max.

Response of the authors: Following the reviewer's suggestions, introduction has been rewritten.

Sentence lines 63-68 is too long and hard to understand. Rewrite

Response of the authors: Following the reviewer's suggestions, those lines has been rewritten.

Line 68-69 / Please add all available research in this field and comment on them in the discussion. Add:

 https://www.researchgate.net/publication/354381111_The_Relationship_between_Anthropometric_Characteristics_and_Motoric_Performance_of_Female_Futsal_Players

 Response of the authors: Following the reviewer's suggestions, the reference has been added and discussed.

Methods

How did you determine the sample size (G*Power or any other method)? Report.

Response of the authors: Thank you for this appreciation. We have completed the Data analysis point including a post-hoc power analysis. A post-hoc power analysis was conducted assuming a confidence level of 0.95. Statistical power is 67% and it was calculated following the Fisterra (https://www.fisterra.com/formacion/metodologia-investigacion/determinacion-tamano-muestral/) sample size guide. It is worth mentioning that the sample is part of the best futsal league in the world, being one of the teams ranked in the top 4 for several consecutive seasons. Therefore, within the elite sport of women's futsal, the sample can be much more representative than if it were an amateur futsal team.

What is their training experience and how many times per week do they train? report

Response of the authors: Following the reviewer's suggestions, information regarding the training has been added into material and methods section.

The sample size is very small and the overall power of your study is extremely low. This is perhaps a solid case study.

Response of the authors: In the elite athlete population it is common to find small samples due to their rareness within the sports world. The vast majority of studies use small samples corresponding to one or several teams from different sports. The amount of accessible sample is usually lower as the total population of elite athletes is smaller compared to non-elite or amateur categories.

  • Bescós Garcia, R.; Rodríguez Guisado, F.. Low Levels of Vitamin D in Professional Basketball Players after Wintertime. Nutricion Hospitalaria. Nutr. Hosp. 2011, 5, 945–951.
  • Anderson, D.E. The Impact of Feedback on Dietary Intake and Body Composition of College Women Volleyball Players over a Competitive Season. J. Strength Cond. Res. 2010, 8, 2220–2226.
  • Mielgo-Ayuso, J.; Zourdos, M.C.; Calleja-González, J.; Urdampilleta, A.; Ostojic, S.M. Dietary Intake Habits and Controlled Training on Body Composition and Strength in Elite Female Volleyball Players during the Season. Appl. Physiol. Nutr. Metab. 2015, 40, 827–834, doi:10.1139/apnm-2015-0100.
  • Nepocatych, S.; Balilionis, G.; O’Neal, E.K. Analysis of Dietary Intake and Body Composition of Female Athletes over a Competitive Season. Montenegrin J. Sport. Sci. Med. 2017, 6, 57–65, doi:10.26773/mjssm.2017.09.008.
  • Waly, M.I.; Kilani, H.A.; Al-Busafi, M.S. Nutritional Practices of Athletes in Oman: A Descriptive Study. Oman Med. J. 2013, 28, 360–364, doi:10.5001/omj.2013.103.
  • Tsoufi, A.; Maraki, M.I.; Dimitrakopoulos, L.; Famisis, K.; Grammatikopoulou, M.G. The Effect of Professional Dietary Counseling: Elite Basketball Players Eat Healthier during Competition Days. J. Sports Med. Phys. Fitness 2017, 57, 1305–1310, doi:10.23736/S0022-4707.16.06469-0.
  • Some authors have highlighted this limitation at a general level in the world of sport as well as the difficulty of extrapolating results.
  • Sands, W.; Cardinale, M.; McNeal, J.; Murray, S.; Sole, C.; Reed, J.; Apostolopoulos, N.; Stone, M. Recommendations for Measurement and Management of an Elite Athlete. Sports 2019, 7, 1–17, doi:10.3390/sports7050105.
  • Atkinson, G. Does Size Matter for Sports Performance Researchers? J. Sports Sci. 2003, 21, 73–74, doi:10.1080/0264041031000071038.

Line 96 - was obtained from participants

Response of the authors: Following the reviewer's suggestions, the sentence has been rewritten

Line 101 - what about previous injuries in like 6 months before testing? This could also influence performance

Response of the authors: Information about previous injuries has been added.

Why were anthropometric measurements taken in the afternoon and not in the morning? 

Response of the authors: Information on anthropometric measurements and instructions for measurement have been indicated following the international standards recommended by the ISAK: Esparza-Ros, F.; Vaquero-Cristóbal, R.; Marfell-Jones, M. Protocolo Internacional Para La Valoración Antropométrica; UCAM Universidad Católica de Murcia, Ed.; Sociedad Internacional para el Avance de la Cineantropometría: Murcia, Spain, 2019; ISBN 978-84-92986-17-0.

The technical error of measurements was 7.5%. Also, how was this measured - you need to report this. Sorry but this is very high/too high for a level 3 Isak accredited anthropometrist? It should be max 5% or less. This means your data is not reliable, therefore all your conclusions are doubtful.

Response of the authors:  According to the reviewer's suggestions, the technical errors of measurement has been indicated for the anthropometric measurements made (0.04% for basic measurements, 2.34% for skinfolds, 0.26% for girths and 0.35% for breadths).

Line 166 - how many repetitions did they make?

The agility test was not done according to the reference used. 10 yards and 5 yards are not equal to 10m and 5m! Therefore you did not perform the same test! This is a scientific paper and not a magazine article so you should be specific! Also when they reach the B point - they don't turn left and run they shuffled to the left which is a different movement. How many times was this tested? Why did you use the stopwatch and not the micro gate system as you used in the RSA test?
Response of the authors:  The authors would like to thank the reviewer for his appreciation.  The authors have used a modified test from the one referenced in the text. The reference has been updated with another study that carried out the same test. We add two references where other authors use the same procedure.

  • Asencio-Vicedo, P., Sabido-Solana, R., García -Valverde, A. & Hernández -Davó, J.L. Relación Entre Distintos Test Con Sobrecarga Excentrica y Test de Cambio de Dirección. Rev. Prep. Fis. en el Fútbol 2019, 10–19.
  • Abián, P.; Del Coso, J.; Salinero, J.J.; Gallo-Salazar, C.; Areces, F.; Ruiz-Vicente, D.; Lara, B.; Soriano, L.; Muñoz, V.; Lorenzo-Capella, I.; et al. Muscle Damage Produced during a Simulated Badminton Match in Competitive Male Players. Res. Sport. Med. 2016, 24, 104–117, doi:10.1080/15438627.2015.1076416.

 The attempts of the T-test (2 attempts) have been added. The stopwatch was not used to measure T-Test, it was measured with the photocells we used for the RSA and Running speed test, authors added this information in their paper.

Lower body power - what was the break between tests? which result was taken to further analysis?

Response of the authors: The authors have added this information in to the manuscript in performance testing section. Break between test was 3 minutes (minimum) and best jump was used to make the analysis.

What was the break between tests and exact order?

Response of the authors: The authors have added the rest time left between tests, 10 minutes, and the exact order in which the tests were performed as well as the distribution over the two sessions. “In the first session the jumping test, running speed test and RSA test were performed in that order. Agility test (T-Test) and aerobic endurance capacity test (“Yo-Yo”) was performed in second session in that order.”

The limitations paragraph is modest and should address error rates, anthropometrical measurements were done in the afternoon and putting together all different playing positions.

Response of the authors: The authors added information related to limitations in anthropometric measurements and playing positions.

On the basis of this, I am rejecting this paper. However, I hope the feedback will be helpful to authors in further studies and in corrections.

Kind regards

Round 2

Reviewer 3 Report

Dear Authors

Thank you for addressing my comments.

I still don't know how have you now come to the lower error rates in anthropometric measurements from the initially reported 7.5%.

In its current condition, the paper is acceptable for publication.

Kind regards